# Identifying the Impact of *Chlamydia trachomatis* Screening and Treatment on Mother-to-Child Transmission, and Respiratory Neonatal Outcomes in Mexico

**DOI:** 10.3390/pathogens13100843

**Published:** 2024-09-28

**Authors:** Gabriel Arteaga-Troncoso, Marcela López-Hurtado, Gabino Yescas-Buendía, María J. de Haro-Cruz, Iván Alejandro Arteaga-Martínez, Jesús Roberto Villagrana-Zesati, Fernando M. Guerra-Infante

**Affiliations:** 1Department of Cellular Biology and Development, Instituto Nacional de Perinatología, Ciudad de Mexico 11000, Mexico; drgarteagat@yahoo.com.mx; 2Military School of Health Officers, Universidad del Ejército y Fuerza Aérea, Secretaría de la Defensa Nacional, Ciudad de Mexico 11650, Mexico; 3Department of Infectology and Immunology, Instituto Nacional de Perinatología, Ciudad de Mexico 11000, Mexico; diaclaro2000@yahoo.com.mx; 4Neonatal Intensive Care Unit, Instituto Nacional de Perinatología, Ciudad de Mexico 11000, Mexico; ybgunam@yahoo.com.mx; 5Department of Veterinary Microbiology, Escuela Nacional de Ciencias Biológicas, Instituto Politécnico Nacional, Ciudad de Mexico 11340, Mexico; deharoc@yahoo.com.mx; 6Department of Morphological Sciences and Human Embryology, Escuela Superior de Medicina, Instituto Politécnico Nacional, Ciudad de Mexico 11340, Mexico; iarteagam2000@alumno.ipn.mx; 7Department of Gynecology and Obstetrics, Instituto Nacional de Perinatología, Ciudad de Mexico 11000, Mexico; dr_robertovillagrana@hotmail.com

**Keywords:** *Chlamydia trachomatis*, prematurity, nucleic acid amplification tests, RDS, hybrid designs in clinical trials, case–cohort design

## Abstract

*Chlamydia trachomatis* (*C. trachomatis*) screening and treatment in pregnancy allows the opportunity to reduce adverse pregnancy and neonatal outcomes worldwide. Although *C. trachomatis* infection is easily treated and cured with antibiotics, only some countries have routine pregnancy screening and treatment programs. We therefore evaluated whether just one maternal screening for *C. trachomatis* is enough to prevent adverse pregnancy and negative neonatal outcomes. Among the 4087 first-time gynecological–obstetric consultations granted at the National Institute of Perinatology in 2018, we selected the study population according to a case–cohort design. Antenatal *C. trachomatis* screening and treatment interventions were performed on 628 pregnant women using COBAS^®^ TaqMan CT. *C. trachomatis* DNA was also detected in samples from 157 infants of these mothers. In the maternal cohort, incidence of *C. trachomatis* infection was 10.5%. The vertical transmission rate was 1.5% for the cohort of mothers who tested positive for *C. trachomatis* and received treatment, and 29.7% for those with a negative test. By evaluating symptomatic neonatal infection, the hazard rate of perinatal pneumonia was 3.6 times higher in *C. trachomatis*-positive babies than in *C. trachomatis*-negative babies. Despite the low rate of mother-to-child transmission in women positive for *C. trachomatis*, possible maternal infection that is not detected in pregnancy significantly increases the risk of neonatal infection with consequent perinatal pneumonia.

## 1. Introduction

The World Health Organization estimated that in 2020, 128.5 million new cases of *C. trachomatis* occurred among adults and adolescents, particularly 76 million (58%) in low-income countries [1,2,3]. Vaginal infection with this sexually transmitted pathogen is asymptomatic in 75% of cases, meaning that patients often do not seek medical treatment until symptoms become substantial. The infection is more common in 15- to 49-year-old women and men, with an estimated global prevalence of 4.2% and 2.7%, respectively [3].

The consequences of untreated maternal STIs, such as *C. trachomatis* and *N. gonorrhoeae* in pregnancy, may affect the well-being of infants born to high-risk groups of HIV-infected women by increasing the possibility of adverse outcomes in these newborns, even with a higher risk than that acquired by HIV infection alone in pregnant women in low- and middle-income countries [4]. In Mexico, a study based on vaginal swab samples found a 6.7% prevalence of *C. trachomatis* in a group of pregnant women; in comparison, a group of more than 1700 infertile women showed a prevalence of 3.5% [5]. The age group between 12 and 19 years had the highest risk, with a vaginal infection rate almost three times higher than the other age groups [5].

Chlamydial vertical transmission rates are relatively high, since 50–70% of babies acquire the infection directly from their untreated mothers. Moreover, 30–50% of *C. trachomatis* cases present conjunctivitis and pneumonia [6,7,8]. In the neonatal stage, it is usually associated with upper respiratory tract infections because the nasopharynx is the most frequently infected site, and approximately 20% of newborns develop early-onset pneumonia and ocular infection [9,10]. Previous studies have documented direct infection associated with damage to the lung, brain, liver, and kidney in premature infants of mothers with a history of vaginal infection [11]. In addition to multisystemic and ocular disease, *C. trachomatis* has also been detected in the peripheral leukocytes of infected newborns [12]. In infants, complete blood cell counts revealed peripheral eosinophilia with increased IgM anti-*C. trachomatis* antibodies, particularly in full-term infants with pneumonitis. However, infected preterm infants did not show higher eosinophil levels than uninfected infants [13].

Failure to recognize symptoms and treat newborns’ underlying cause of respiratory distress can lead to short- and long-term complications, including chronic lung disease, respiratory failure, and even death. Alveolar surfactant deficiency, which increases alveoli surface tension, results in microatelectasis and low lung volumes, and exposure to *C. trachomatis* at an earlier gestational age frequently leads to more serious early and late respiratory infections in the baby [14,15]. In some countries, screening of pregnant women for *C. trachomatis* infection is recommended to prevent infection of newborns. Since the transmission of the maternal disease to their babies can occur in utero, routine screening and retesting must be carried out to prevent contagion during pregnancy and avoid harm to the babies [16].

The strength of the evidence has been limited by studies directly comparing pregnancy or perinatal outcomes between treated and untreated *C. trachomatis*-infected mothers. To abolish the major source of bias due to loss to the follow-up of pregnant women, the case–cohort design was useful for analyzing the time to mother-to-child transmission event (failure event) in a large cohort in which failure rarely occurs. Therefore, in the present study, we analyzed two comparative cohorts of mothers and their babies from an initial large pool of pregnant women to identify risk sets throughout different times of maternal *C. trachomatis* screening using nucleic acid amplification. Our study aims to question whether just one maternal screening of *C. trachomatis* is enough to prevent adverse pregnancy and negative neonatal outcomes.

## 2. Materials and Methods

### 2.1. Ethics Statement

The Ethics Committee of the National Institute of Perinatology “Isidro Espinosa de Los Reyes” (INPer, CONBIOÉTICA-09-CEI-021-20170823) approved this research, following the ethical standards established in the Declaration of Helsinki. All participating women were adequately informed about the study’s objectives, stating their consent to participate by signing the written informed permission. Data were obtained of analysis of endocervical samples from pregnant women, and samples of rhinopharyngeal secretion, tracheal lavage, conjunctival secretion, and/or peripheral blood from their newborns were obtained for the evaluation of mother-to-child transmission of *C. trachomatis*. These evaluations were conducted following PRISMA-IPD guidelines for preferred reporting items in systematic reviews and meta-analyses of individual participant data [17].

### 2.2. Study Design

The present study was carried out in department of gynecology and obstetrics and neonatal care unit of the National Institute of Perinatology “Isidro Espinosa de Los Reyes” (INPer) in Mexico City from 1 January to 31 December 2018. A total of 628 pregnant women was randomly selected for vaginal screening for *C. trachomatis* infection at the first prenatal visit between 7 and 38 weeks of gestation. Two swabs of endocervical samples were collected for microbiology analyses and the identification of *C. trachomatis* DNA. Each endocervical sample was obtained before the administration of any medical treatment, and the patients were excluded from the investigation due to various circumstances, such as (a) the application of vaginal douches before taking the endocervical sample; (b) systemic or local antimicrobial therapy 30 days before evaluation; and (c) sexual intercourse 48 h before sample collection. Vaginal swabs were collected independently of gestational age, and only women who were positive received a single 1 g oral dose of azithromycin [18]. The test-of-cure was not repeated after providing antimicrobial therapy during pregnancy. Laboratory diagnosis of vaginal infections was performed routinely and described elsewhere [19].

Data on HPV and HIV were collected from the patients’ medical records. Pregnant women with a history of HPV infection with/without clinical features of external genital warts were referred to the colposcopy section for the visualization of temporary aceto-white lesions. Biopsies of the affected tissue were obtained for analysis by the histopathology and DNA of HPV genotyping (Linear Array HPV Genotyping Test in vitro^®^, Roche diagnostics, Pleasanton, CA, USA), respectively [20]. HIV infections were diagnosed by virologic tests that detected viral RNA load (RealTime HIV-1 viral load assay, Abbott Molecular, Des Plaines, IL, USA). Automated extraction of viral nucleic acid was performed using purification reagents, and processing and testing on plasma samples was performed following the manufacturer’s instructions. Quantitative assay results were reported as copies/mL [21]. Diagnosis of chronic hepatitis B (HBV) infection was based on the detection of hepatitis B surface antigen (HBsAg) in blood (Abbott Architect HBsAg Qualitative II assay; Abbott Diagnostics, Wiesbaden, Germany), according to the manufacturer’s recommendations. The sensitivity of HBsAg assay calculated for diagnosing HBV in humans is 99.8% (99.4–99.96%) and analytical sensitivity 0.011 IU/mL [22]. Mothers who tested positive for HPV, HIV, and/or bacterial vaginosis were treated accordingly. All patients received the same indications in their pregnancy follow-up, as indicated by our medical protocols, including those positive for HIV. Adherence to postmaternal antiretro-viral therapy of HIV-positive pregnant mothers who were eligible for this study was carried out in other hospitals.

A sample of mothers who were *C. trachomatis*-test positive and their offspring were considered an eligible cohort, and the mothers who tested negative were a subcohort or control group. Thus, two comparative groups of mothers were created, generating infant risk sets for each maternal *C. trachomatis* screening. The risk sets were formed according to the timing of maternal screening, stratifying them into three categories based on properties of the normal distribution: (a) babies from mothers who were diagnosed before 16 weeks of gestation, “early antenatal detection”; (b) babies from mothers who were diagnosed between 16.1 and 29 weeks of gestation, “mid-antenatal detection”; and (c) babies from mothers who were diagnosed after 29.1 weeks of gestation, “late antenatal detection”. The case–cohort design included all cases of *C. trachomatis*-positive babies that occurred in the cohort, and controls were randomly selected from those newborns available of the subcohort at that time and matched according to the maternal screening categories created by the COBAS^®^ TaqMan CT test (Figure 1).

Blood samples from mothers and their babies were collected as part of the medical protocol of our hospital. The newborns were diagnosed with *C. trachomatis*-induced respiratory distress based on clinical, radiological, and/or molecular studies. Importantly, regardless of neonatal infection, all babies born prematurely or with low birthweight were admitted to neonatal intensive care unit, where a case report form was completed for each baby. At the same time, the following samples were obtained from each baby: (1) rhinopharyngeal secretion for the isolation of atypical microorganisms; (2) tracheal lavage (if the baby required intubation) for the detection of *C. trachomatis*; (3) conjunctival discharge (if the baby showed eye redness, eyelid swelling, and/or discharge); and (4) a peripheral blood sample for determining cell count and C–reactive protein concentration and *C. trachomatis* infection. Complementary data were obtained by direct interviews with the mothers, and consultation of the mother and her baby’s medical records. Data corresponding to newborns who died within 72 h of life were excluded from the analysis.

### 2.3. Definitions of Other Variables

Vaginal infections in the mother and perinatal complications such as prematurity and low birthweight were evaluated. Bacterial vaginosis was defined as a syndrome characterized by one or more of following signs and symptoms: discharge, itching, burning, irritation, dysuria, dyspareunia, and vaginal fetidity, secondary to the presence of *Gardnerella vaginalis* (*G. vaginalis*). Preterm birth was defined as a birth occurring before 37 weeks of gestation, and the babies were considered growth-restricted using the proxy of “Small for gestational age” definition, that is, infants at or below the 10th percentile of birthweight-for-gestational age based on the quadratic function of boys and girls separately. To assess the relationship between genital tract infection and preterm birth, babies were classified as “full term” if they were born after 37 weeks of gestation, “preterm” if the delivery occurred between 36 and 37 weeks of gestation, “moderate to late preterm” if the delivery occurred between 32 and 36 weeks of gestation, and “very preterm” if the delivery occurred before 32 weeks of gestation. Spontaneous preterm delivery was considered if birth occurred before 34 weeks of gestation.

All infants were diagnosed with neonatal respiratory distress syndrome with clinical, laboratory, and radiological criteria. Upon clinical examination, such babies had signs and symptoms of increased work of breathing, including nasal flaring, expiratory whimper, xiphoid retraction, intercostal drawing, thoracoabdominal dissociation, polypnea, exhaustion, and apnea. Laboratory criteria included hypoxia, hypercapnia, asphyxia, acidosis, the presence of PaO2 < 50 mmHg, central cyanosis breathing room air, or the need for supplemental oxygen to maintain PaO2 > 50 mmHg. Radiographic criteria corresponded to those typical of the clinical picture, such as diffuse reticulo-granular image with air bronchogram or ground glass image. The causes of neonatal respiratory distress syndrome, including perinatal pneumonia (PP), respiratory distress syndrome (RDS), transient tachypnea of the newborn (TTN), and meconium aspiration syndrome (MAS), were considered in the differential diagnosis of respiratory distress in the newborn [23]. Postpartum complications, including conjunctivitis and sepsis, were identified within 72 h after birth.

### 2.4. Diagnosing C. trachomatis Infection

A calcium alginate swab was introduced into the vaginal cavity and gently scraped in the cervical area. The swabs were placed in transport media included in the commercial kit for *C. trachomatis* detection, COBAS^®^ TaqMan CT Test v2.0 (Roche Molecular System, Oklahoma City, OK, USA). The advantage of NAAT is that both the cryptic plasmid and the *ompA* gene targets can be detected independently, increasing the sensitivity for different *C. trachomatis* serovars [24]. The sensitivity and specificity of the COBAS^®^ TaqMan CT Test on clinical specimens are 93% and 99.8%, respectively [25].

### 2.5. Detection of C. trachomatis in Babies

Peripheral blood, conjunctive, and/or bronchial aspirate samples from each newborn with respiratory distress were evaluated. Neonatal *C. trachomatis* cases were considered positive when at least one of the samples obtained was positive. To increase the diagnostic sensitivity of PCR and confidence in negative results, specimens were collected from babies with respiratory infections, and DNA was obtained by the phenol–chloroform method followed by precipitation with cetyltrimethylammonium bromide (CTAB) [26]. Positive *C. trachomatis* detections were determined by endpoint PCR amplification of the 129 bp fragment of the *C. trachomatis ompA* gene. The primers and the reaction conditions used in the assays were performed as previously described by López-Hurtado et al. [12,13]. The sensitivity and specificity of in-house PCR assay calculated for diagnosing *C. trachomatis* in blood sample from newborns are 94.4% and 88.5%, while bronchoalveolar lavage samples 27.8% and 92.3%, respectively [27].

### 2.6. Statistical Analyses

Statistical analyses were performed with IBM SPSS v25 software (IBM Corp: Armonk, NY, USA). In all cases, the analysis (two-sided test) was considered statistically significant at *p* < 0.05. For continuous variables, a statistically significant difference was determined using Student’s *t*-test or the Mann–Whitney U test in the case of nonnormality. Categorical data were evaluated using Fisher’s exact test, and correlations between categorical and continuous variables were examined using Spearman’s rho. The Kolmogorov–Smirnov test was used to verify data normality. We analyzed time-to-event data under a case–cohort design using the Kaplan–Meier method. The survival analysis evaluated the distribution of time-to-event of maternal screening of *C. trachomatis* infection by comparing mother-to-child transmission between mothers diagnosed as positive and negative at different moments in pregnancy with the log-rank Mantel–Cox test. The proportional hazards Cox model was adjusted to test the hypothesis of specific factors influencing the hazard rate of mother-to-child transmission, perinatal pneumonia, respiratory distress syndrome, or transient tachypnea, stratifying by birthweight as a covariate. For multivariable analysis, the selection of variables was stepwise forward with *p* < 0.05 for integrating this parameter in the model, excluding all those that did not meet the selection criteria from the Cox model. The proportional hazards assumption was evaluated using the lines present in the hazard curve plot, which provides the visual representation of the corresponding hazard rate of groups that do not intersect.

Attributable risk, expressed as a population attributable fraction (AR), was calculated to assess the portion of disease burden preventable by avoiding exposure [28,29] using the following equation:(1)$$AR=1-\sum_{j=1}^{n}\frac{p_{j}}{RR_{j}}$$
where *j* = 1, 2, …, n are the levels of exposure; *p_j_*: proportion of *C. trachomatis*-positive babies at *j*th exposure level; *RR_j_*: adjusted relative risk at level j estimated by the Cochran–Mantel–Haenszel method.

## 3. Results

### 3.1. Maternal Characteristics and Obstetric Consequences

The patient population studied included a total of 628 pregnant women with a median age of 22 years (16–31) and 157 newborns. Table 1 summarizes the demographic and clinical characteristics of pregnant women. Patients presented at a median gestational age of 21 weeks and 1 day (15.1–28); 8.9% of them presented with premature labor, and 13.9% after spontaneous rupture of membranes (with or without contractions). Overall, 66 women (10.5%) tested positive for *C. trachomatis* as one of several vaginal infections, and of these, 53 women (8.4%) were *C. trachomatis* positive only in endocervical specimens (all of them were subsequently treated with azithromycin). After adjusting for gestational age at premature rupture of membrane (PROM), the women who received antibiotic prophylaxis after 16 weeks gestation had a 2.9 times higher risk of PROM than women who received treatment before that time (RR: 2.9; 95% CI 1.5–5.8; Mantel–Haenszel test, *p* < 0.002). The maternal infection rate was significantly higher in adolescent pregnancies (16.2%) than in adult pregnancies (6.2%) (RR: 2.6; 95% CI 1.6–4.3; *p* < 0.001).

The overall prevalence of *C. trachomatis* infection in women with history of HPV and HIV infection in this study was 12.1% and 1.5%, respectively. The *C. trachomatis* infection associated with history of HPV infection only (RR: 2.7 95% CI 1.4–5.1, *p* < 0.01). In this group of patients, only two (2/31, 6.5%) pregnant women showed clinical features as anogenital warts around the vulva lips, vagina, or cervix. Of the 31 patients undergoing colposcopy, 17 (54.8%) pregnant women did not show clinical lesions compatible with HPV infection, and 14 (45.2%) had colposcopic findings of LSIL confirmed by histopathology. A biopsy was performed in these women, where two (2/14, 14.3%) showed high-risk genotypes (HPV-39 and -71 genotypes), and multiple-type infection was only found in only one (7.1%) of the HPV-positive women (genotypes HPV59/70).

Only one (1/66, 1.5%) mother among the *C. trachomatis*-positive pregnant women was an HIV patient, and 17 mothers (3%) among the *C. trachomatis*-negative pregnant women were HIV patients. The overall incidence of HIV infection in this population was 2.9%. Of the 18 HIV-pregnant women, four (4/18, 22.2%) patients had HIV-1 RNA > 400 copies/mL, and CD4 T cell counts had a median 603 cells/µL (IQR 222.8–744), and the median CD4/CD8 ratio was 0.5 (IQR 0.14–0.85).

On the other hand, the recovery rates of *G. vaginalis*, *U. urealyticum* and *S. agalactiae* (Group B streptococcus) in endocervical samples from *C. trachomatis*-positive pregnant women are shown in Figure 2. Both for women with *C. trachomatis* positive detection and negative women, there were no significant proportional differences in the rates of isolation of the three studied organisms from the endocervical samples. All organisms were, however, isolated less frequently from the *C. trachomatis* positive women, in whom the isolation rates were approximately the same as those in the adults and the teenagers. *G. vaginalis* was significantly associated with *C. trachomatis* (RR: 6.4; 95% CI 4–10.2, *p* < 0.001) and was less common in women who tested in the first and second (5/480, 1%) compared to the third trimester of pregnancy (4/148, 2.7%).

### 3.2. Characteristics of the Babies after Screening and Treatment of Their Mothers

Overall, 157 samples from newborns (123 peripheral blood, 18 nasopharyngeal, and 16 conjunctival swabs) were analyzed. In the included cohort, the incidence of neonatal *C. trachomatis* infection was 21% (n = 33). Twelve babies with peripheral blood positive samples alone, six babies with nasopharyngeal samples, eight babies with conjunctival samples, and two babies with mixed sample detection were identified (Spearman correlation 0.32; *p* < 0.01). Table 2 compares the common neonatal outcomes of newborns with positive or negative tests for *C. trachomatis*, as well as their frequencies of perinatal pneumonia, respiratory distress syndrome (RDS), conjunctivitis, and sepsis. The percentage of *C. trachomatis*-positive babies was inversely correlated with gestational age among mothers who were negative for the COBAS TaqMan CT test (Spearman’s correlation −0.24, *p* < 0.03). Preterm infants were most frequently infected with chlamydia (22.5%; 20/89); for reference, only approximately 11.8% (8/68) of full-term infants were associated with the infection. Likewise, infected infants had a relative risk of extremely low birth weight of 2.5 (95% CI 1.3–5; *p* < 0.008) compared with uninfected infants, whereas the low birth weight of infected infants was not different from that of uninfected infants (RR: 2.12; 95% CI 0.99–4.52, *p* < 0.059). The cases and noncases differed greatly in their outcomes of perinatal pneumonia (*p* < 0.001), conjunctivitis (*p* < 0.02), and late-onset sepsis (*p* < 0.008). Only one positive culture (3.5%) of the total positive neonatal samples was detected in the full-term baby from a mother with a *C. trachomatis*-positive test.

### 3.3. Analyses of the Effectiveness of Antenatal Screening and Treatment

To estimate the probability of neonatal *C. trachomatis* infection, 157 births were included and classified: 66 newborns from *C. trachomatis*-positive mothers and 91 newborns of the random subcohort of negative mothers (Figure 3). Based on the Kaplan–Meier analysis at the time of pregnancy, the estimated proportion of infected babies was 1.5% when the mother was treated (95% CI, 0–4.54%) and 29.7% when the mother was untreated (95% CI, 20.1–39.2%). The estimated absolute difference in infection between the two groups of babies was 28.2% (95% CI, 16.7–39.6%; Z = 5.61; *p* < 0.001). The test of equality of survival distributions for the different levels of antenatal *C. trachomatis* screening during 40 gestational weeks was significant (Log-rank Mantel–Cox = 24.03, *p* < 0.001).

We considered five potential Cox proportional hazards models that included mother-to-child transmission, perinatal pneumonia, RDS, conjunctivitis, and late-onset sepsis. Explanatory variables were adjusted for maternal age, smoking, alcohol consumption, and sex of newborns (Table 3). All models satisfy the proportional hazard assumption of the Cox model, and the plot of hazard curves shows the effect of birth weight stratification.

Only two of the analyzed risk models showed the effect of neonatal *C. trachomatis* infection as a covariate; these models corresponded to mother-to-child transmission and perinatal pneumonia (Figure 4). The risk of mother-to-child transmission among mothers with a maternal *C. trachomatis*-negative test was increased 24.4-fold in contrast to mothers with a positive screen who received treatment (HR: 24.4; 95% CI 3.1–194.3; *p* < 0.003), and the risk of mother-to-child transmission differed significantly in birthweights under 1500 g (HR: 4.4: 95% CI 1.7–11.4: *p* < 0.003) (Figure 4a).

Likewise, the estimated risk of perinatal pneumonia in chlamydia-positive babies was 3.6 times higher than that in *C. trachomatis*-negative infants (HR: 3.55, 95% CI: 1.48–8.51, *p* < 0.004). The risk of perinatal pneumonia estimated using birthweight stratification showed that the “high risk of pneumonia” in *C. trachomatis*-infected babies differed significantly in birthweights between 1500 and 2500 g (HR: 17.29; 95% CI 1.91–156.6, *p* < 0.01) and lower than 1500 g (HR: 83.71; 95% CI 9.95–8.51, *p* < 0.001) (Figure 4b). The natural interpretation of this plot is that the risk of maternal–child transmission and perinatal pneumonia for babies under 2500 g increases over gestational time since the gradient/slope of the cumulative hazard function also increases over time. Regardless of the infection in the neonate, we found the relative risks of respiratory distress associated with moderate to late preterm (RR 1.7; 95% CI 1.4–4; Mantel–Haenszel test, *p* < 0.02) and premature deliveries (RR 1.8; 95% CI 1.34–4.12), especially by cesarean section.

Table 4 summarizes data regarding adjusted estimates of the relative and attributable risks of mother-to-child transmission and symptomatic infection from *C. trachomatis*-positive newborns by maternal screening categories during pregnancy. There was a statistically significant trend in the strength of the association between mother-to-child transmission (*p* < 0.001), neonatal *C. trachomatis* infection, and the symptomatic onset of perinatal pneumonia (*p* < 0.002), conjunctivitis (*p* < 0.007), or late-onset sepsis (*p* < 0.01) after stratifying by maternal screening time. The relative risk adjustment obtained by the Cochran–Mantel–Haenszel method was consistent with the Cox regression model for mother-to-child transmission and perinatal pneumonia only but not for conjunctivitis or late-onset sepsis. The population attributable fraction showed a 97.7% (95% CI, 96.5–100%) relative decrease in the risk of mother-to-child transmission if maternal chlamydial infection was eliminated. Additionally, a 65% (95% CI, 47.3–82.7%) reduction in the risk of perinatal pneumonia in *C. trachomatis*-positive infants could be prevented if the maternal exposure factor was removed from the population.

## 4. Discussion

This study was the first to document the time-to-event of maternal *C. trachomatis* infection screening in mother-to-child transmission in a low-prevalence setting. The most important point to prevent prenatal *C. trachomatis* infection is a methodological strategy for the screening and treatment of all pregnant women for the microorganism, regardless of symptoms. It appears that a single maternal *C. trachomatis* screening test is not sufficient to prevent adverse pregnancies and negative neonatal outcomes. Upon screening for *C. trachomatis* with a maternal positivity of 10.5% (n = 66) detected by COBAS^®^ TaqMan CT Test, remission of the infection was verified in 28 newborns. The lack of *C. trachomatis* detection in 27 mothers who had *C. trachomatis*-positive children was one of our most interesting results. Our finding that 29.7% of babies from *C. trachomatis*-negative mothers showed neonatal *C. trachomatis* infection suggests a failure to detect maternal infection.

The failure in maternal diagnosis also could be related to the site of *C. trachomatis* colonization at the time of collected of endocervical sample, as well as to the low number of elementary or reticular bodies in the sample swab, inhibitors of gene amplification target, presence of different *C. trachomatis* variants, or screening test before acquiring the chlamydial infection. Disruption of vaginal homeostasis could facilitate the migration of *C. trachomatis* to the upper genital tract, being undetected by the COBAS^®^ TaqMan CT test [30]. Herein, we identified the presence of different bacterial communities in the endocervix with tissue changes in the vagina from pregnant women who tested negative by the COBAS TaqMan CT test, suggesting the possible migration of *C. trachomatis* to the upper genital tract. Any changes in vaginal balance affect the mucosa integrity that is functional to the outside and inside permeability gradient [31]. Remodeling of the vaginal mucosa layer that continues to the uterus becomes an easy site for the colonization of infectious agents and inflammatory processes that accelerate the deterioration process of both the endothelial wall and the microbiota shield, thus facilitating the accumulation of proinflammatory proteins and long-term pathogens [32].

Another explanation for the high percentage of *C. trachomatis*-positive babies among their NAAT–negative mothers is that mothers of infected babies acquired a recent strain of *C. trachomatis* from their original partner or a new sexual partner who, in turn, had acquired a new strain from a different pair [33]. However, this possibility cannot be fully supported by our findings, since mothers were screened for *C. trachomatis* after 10–12 weeks of pregnancy, and many infected babies of negative mothers were found. In addition, a fraction of babies infected with *C. trachomatis* (13/27, 48.2%) were born by cesarean section, so they were not infected through the birth canal. This observation suggests that these microorganisms can directly attack the fetus through uterine transmission (in which *C. trachomatis* infects trophoblastic cells) or by aspiration of PROM-contaminated amniotic fluid.

Control strategies that emphasize diagnosis, detection, and effective treatment will lead to a significant reduction in the incidence of maternal *C. trachomatis* infections and an eventual reduction in postnatal complications and neonatal adverse events such as low birthweight and prematurity. Recommendations to include pregnant women in routine screening for infection are currently issued by public health programs and vary from country to country. Failure to enforce these recommendations can lead to poor interventions by the health system to protect and improve the quality of prenatal care to significantly reduce the risk of mother-to-child transmission of infection, particularly in adolescent mothers [34]. The NHS England and the UK NSC recommend that systematic population screening in pregnancy is offered and advised to all eligible women for HIV, hepatitis B, syphilis, and *C. trachomatis*. These provisions only apply to sexually active women and other people with a uterus or ovaries, as well as transgender men and non-binary people assigned female at birth and intersex people with a uterus or ovaries under the age of 25 [35,36]. The United States’, for example, current CDC guidelines recommend that women under the age of 25 and those at increased risk for *C. trachomatis* (i.e., those with a new sexual partner, more than one sexual partner, concurrent partners, or a sexual partner who has an STI) should be screened at the first prenatal visit with re-examination during the third trimester to prevent postnatal maternal complications and infection of the newborn [37]. In Canada, the Public Health Agency recommends screening at the first prenatal visit and re-examination in the third trimester for pregnant women who are positive at the initial test regardless of their age or those who are exposed to the ongoing risk of infection [38]. Based on our study design and data analyses in a low-prevalence population, this evidence supports *C. trachomatis* screening at the first antenatal visit and retesting in the third trimester of pregnancy in all pregnant women, not only those in high-risk groups. The Mexican public health system, although it is one of the relatively stronger health systems and has the greatest development of public health and social security in Latin America, is also, unfortunately, a health system that presents significant vulnerabilities in terms of inequality and inequity in health access. Current regulations in Mexico only consider mandatory notification of *C. trachomatis* infection when cases of lymphogranuloma venereum occur; therefore, routine test results are not compulsory [39]. In our hospital, this situation complicates the implementation of the current CDC guidelines.

To date, randomized clinical trials have not been conducted to assess whether universal or exclusive *C. trachomatis* screening of the high-risk population during pregnancy reduces the mother-to-child transmission rate. This may be due to the harmful nature of *C. trachomatis* and its new variants among sexually active adolescents and young adults and their newborns, which makes any randomized study ethically difficult to conduct. Many systematic reviews have evaluated existing randomized trials with screening tests and antibiotic regimens for the treatment of *C. trachomatis* in pregnancy and the reduction of low birthweight and prematurity [40,41]. While some of the studies have included a placebo or no-treatment control group, they have lacked control groups without *C. trachomatis* infection. An important aspect of our study was the application of a case–cohort design (or, equivalently, incidence density study designs) that identified the outcomes of *C. trachomatis* detection in a cohort of pregnant women in contrast to data drawn from a negative subcohort. This design allowed us to form two sets of infants at risk at each time point in pregnancy: 66 babies of *C. trachomatis*-positive mothers and 91 of mothers who were negative for infection at the same time during pregnancy [42]. For the analysis, the efficacy of early screening and cure of maternal infection were assessed by comparing the proportions of infected babies from a cohort of mothers who tested positive and a negative subcohort at different times during pregnancy. We also identified mother-to-child transmission of *C. trachomatis* and the appearance of perinatal pneumonia in these infected babies. Upon efficacy analysis, the results indicated that there may have been other chlamydia variants in the samples from the mothers undetected by the COBAS TaqMan CT test, which have not yet been identified in Mexico and may favor the expansion of new chlamydia clones. However, the detection of these new variants could only be carried out with high diagnostic coverage in more than 2339 samples [43].

Adverse pregnancy outcomes associated with maternal infections, such as low birthweight and prematurity, may occur due to direct infections of the baby or infections causing premature delivery without directly affecting the newborn. In addition to maternal *C. trachomatis* infection, similar adverse neonatal outcomes have been reported for some pathogens, such as *N. gonorrhoeae* [44,45], but in our statistical analyses after adjusting for the frequency of STIs or other vaginal infections, the effects of these pathogens were not significant, and they were not included in the final models. Importantly, this study documented the possible mechanism of mother-to-child transmission of *C. trachomatis*, suggesting that the microorganisms can directly attack the fetus at an early gestational age or that transmission that can occur within the uterus through a transplacental or ascending infection when the organisms can be found in the space between the decidua and the membranes, within the fetal membranes, in the amniotic fluid, or within the placenta. The respiratory tract and eyes are the main target sites of *C. trachomatis* infection in babies responsible for the clinical manifestations [8,10]. In agreement with this, this study documented the association between neonatal *C. trachomatis* infection and perinatal pneumonia but not conjunctivitis based on proportional hazards Cox analysis.

Another important finding is the biphasic clinical picture of neonatal respiratory distress in *C. trachomatis*-infected newborns. Of the 28 infected newborns, 19 (67.9%) showed respiratory distress characterized by an early onset after birth and a biphasic course. The isolation of *C. trachomatis* in babies and the favorable response to specific antibiotics confirmed the etiologic agent in 17 infants who developed perinatal pneumonia and 2 with RDS, which required assisted mechanical ventilation and supplemental oxygen. Among *C. trachomatis*-negative infants, only 2.3% (3/129) of the cases had mild to moderate transient tachypnea within 4 h of birth, which recovered spontaneously within 2–3 days. Notably, none of the babies requiring complementary therapies died during the CPAP maneuver. Our results provide evidence that neonatal *C. trachomatis* infections may be related to the first phase of respiratory distress, according to Numazaki et al. [46] and Sollecito et al. [47]. This seems to contradict previous reports, since infection by *C. trachomatis* in newborns is generally reported at the end of the neonatal period or beyond and not in the first day of life [7,48]. This suggests that proper management of infants in the first hours of life is crucial.

A limitation of our study was the small number of women with single pathogen infections in the genital tract. This small number was initially based on the aim to pool multiple STIs, including *C. trachomatis*, without considering the specific infection exposure of infants. The ability to detect differences in perinatal outcomes by STI group may have been limited, particularly when exposed to multiple maternal infections combined for analysis of neonatal *C. trachomatis* status. Likewise, although all serious events and clinical results that occurred during the main evaluation of the newborns were registered, our analysis was only limited to the results of neonatal *C. trachomatis* associated with respiratory distress and not to infection by other pathogens such as respiratory syncytial virus, which can result in serious respiratory infections in babies.

## 5. Postpartum Perspectives for Chlamydia-Positive Women and Mothers Who Had *C. trachomatis*-Positive Newborns

Postpartum infections, in general, show a significant social burden because they increase episodes of maternal anxiety and risk of postpartum depression, conditions that interfere with the mother–baby bond and have a negative impact on breastfeeding [49,50].

The results of the study show future implications for medical care during pregnancy and the immediate postpartum period.

All mothers underwent routine testing using COBAS^®^ TaqMan CT for *C. trachomatis* infection only once in perinatal period, upon entry into prenatal care, but did not undergo a second diagnostic test thereafter of treatment during pregnancy or in postpartum period. The lack of re-screening tests, and the possibility of pathogens ascending to the upper genital tract, highlight these women who have contracted *C. trachomatis* infection as a high-risk group after childbirth. Both pregnant women who have been diagnosed with *C. trachomatis* vaginal infection and mothers who have had positive children may be particularly vulnerable to contracting chlamydia infection or another STD after childbirth. The infections that are undetected and inadequately treated during pregnancy have a detrimental effect on women and their babies’ health on the future [51].

## 6. Conclusions

The information contained in this publication is intended for reducing the risk of vertical transmission of *C. trachomatis*. If *C. trachomatis* infection is not detected in mothers, they will not be treated, having a detrimental effect on women and their babies’ health. The recommendation outlined in this study highlights the need to carefully weigh the risks and benefits of early detection and retesting of women with vaginal *C. trachomatis* infection in pregnancy. Those pregnant women should be retested in the third trimester. Because of the high likelihood of mother-to-child transmission, we recommend testing all patients as early as 12 weeks of gestation and highlighting the need for a second screening test in the third trimester of pregnancy, regardless of the treatment status of their partners.

## Figures and Tables

**Figure 1 pathogens-13-00843-f001:**
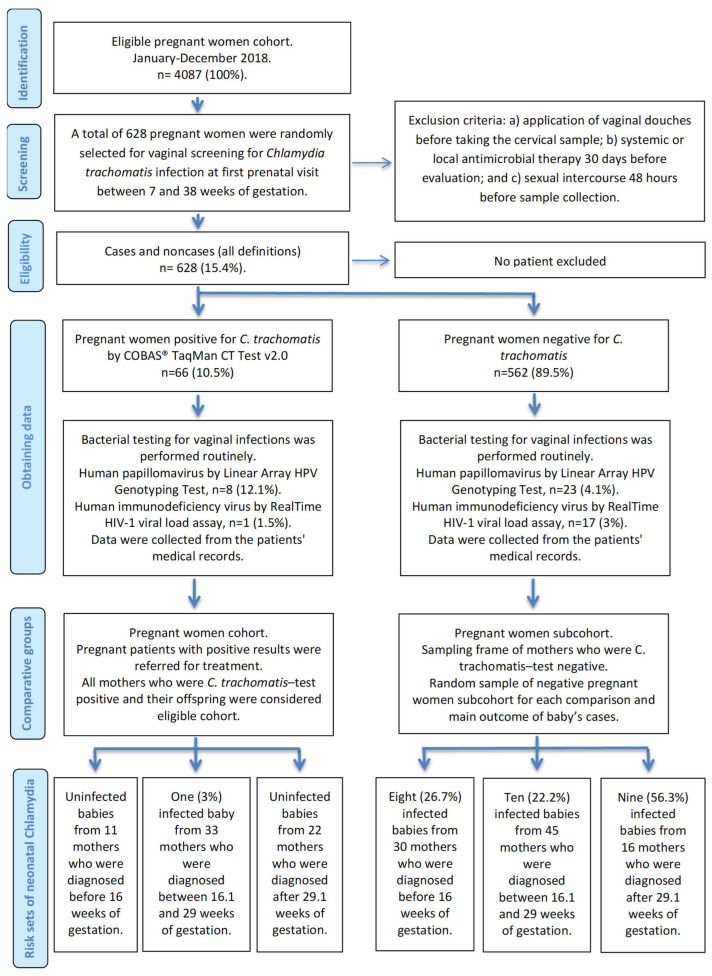
Flow diagram for *Chlamydia trachomatis* screening and treatment in pregnancy, and mother-to-child transmission.

**Figure 2 pathogens-13-00843-f002:**
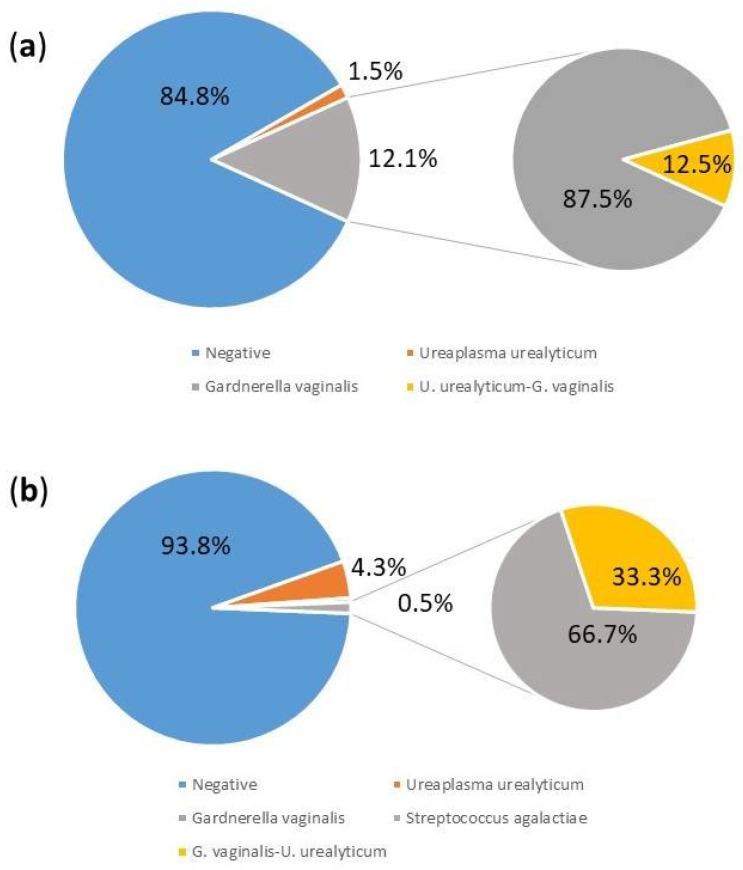
The recovery rates of *G. vaginalis*, *U. urealyticum* and *S. agalactiae* (Group B streptococcus) in endocervical samples from *C. trachomatis*-positive pregnant women (n = 10) (**a**); and *C. trachomatis*-negative (n = 35) included in the study (**b**).

**Figure 3 pathogens-13-00843-f003:**
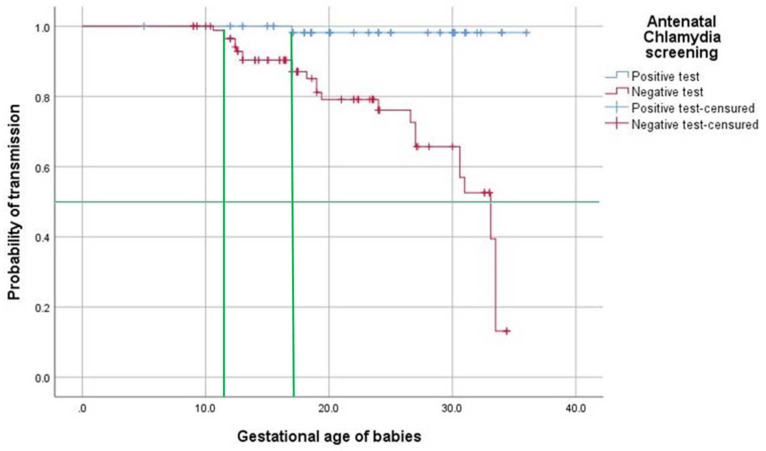
A Kaplan–Meier curve of two baby risk sets. Group A arm: babies from mother positive-test (line blue); group B arm: babies from mother negative-test (line red). Each horizontal small segment is the screening *C. trachomatis* interval in the mother-to-child transmission event between one and the next subject in that arm. Only the event influences the interval length, whereas tick marks indicate censored subjects. Median survival from experiencing the evaluated event could be estimated in both arms by drawing a line on the *y*-axis at 0.5 or median survival at 50% (green line). Located the point at which each intersects, 0.5 shows median survival with a probability of 0.988 of approximately 12 gestational weeks in cohort B and 17 gestational weeks, with probability of 0.982 in cohort A (two perpendicular green lines).

**Figure 4 pathogens-13-00843-f004:**
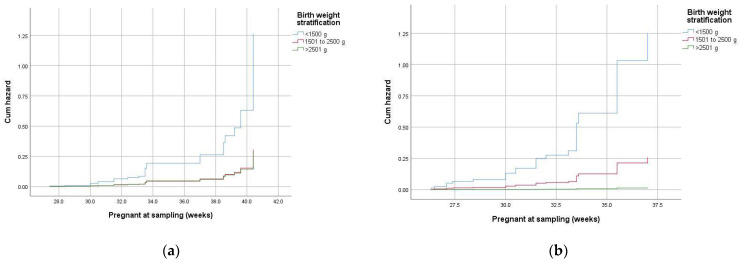
The hazard curves of mother–child transmission risk with respect to birthweight stratification of *C. trachomatis*-positive babies (**a**), and those who develop perinatal pneumonia (**b**).

**Table 1 pathogens-13-00843-t001:** Characteristics of pregnant women, including maternal features at enrollment, clinical history, and obstetric outcomes.

	All Pregnant Women [n (%), n = 628]	Positive Pregnant Women [n (%), n = 66]	Negative Pregnant Women [n (%), n = 562]	*p* Value
**MATERNAL FEATURES**				
Maternal age at delivery—years ± SEM	23.8 ± 0.3	20.5 ± 7.5	24.2 ± 8.2	<0.001
Gestational age at delivery—weeks ± SEM	37.1 ± 0.1	38.03 ± 2.5	37 ± 3.3	0.02
Pregnancy in teenagers	272 (43.3)	44 (66.7)	228 (40.6)	0.001
Gestational age (weeks) * at first maternal visit				N.S. ^1^
<16	168 (26.8)	16 (24.2)	152 (27)	
16.1 to 29	312 (49.7)	28 (42.4)	284 (50.5)	
>29.1	148 (23.6)	22 (33.3)	126 (22.4)	
**CLINICAL HISTORY**				
Recurrent pregnancy loss	13 (2.1)	1 (1.5)	12 (2.1)	N.S. ^1^
Previous abortion	46 (7.3)	6 (9.1)	40 (7.1)	N.S. ^1^
Tubal infertility	69 (11)	7 (10.6)	62 (11)	N.S. ^1^
Human papillomavirus infection	31 (4.9)	8 (12.1)	23 (4.1)	0.01
High-risk genotypes (HPV39 and HPV59)	2 (0.3)	1 (1.5)	1 (0.2)	N.S. ^1^
Low-risk genotypes (HPV6, HPV70, and HPV71)	3 (0.5)	0 (0)	3 (0.5)	N.S. ^1^
Human immunodeficiency virus	18 (2.9)	1 (1.5)	17 (3)	N.S. ^1^
Chronic hepatitis B infection	4 (0.6)	1 (1.5)	3 (0.5)	N.S. ^1^
Endocrinological involvement **	50 (8)	1 (1.5)	49 (8.7)	0.05
Psychiatric disorders ***	30 (4.8)	2 (3)	28 (5)	N.S. ^1^
Smoked cigarettes/Drug addiction	23 (3.7)	2 (3)	21 (3.7)	N.S. ^1^
**OBSTETRIC OUTCOMES**				
Premature rupture of membrane	56 (8.9)	13 (19.7)	43 (7.7)	0.004
Spontaneous preterm delivery	87 (13.9)	5 (7.6)	82 (14.6)	N.S. ^1^
*Mode of delivery*				
Cesarean section	332 (52.9)	35 (53)	297 (52.8)	N.S. ^1^
Vaginal	296 (47.1)	31 (47)	265 (47.2)	N.S. ^1^
Preeclampsia	27 (4.3)	1 (1.5)	26 (4.6)	N.S. ^1^
Neonatal mortality	12 (1.9)	0 (0)	12 (2.1)	N.S. ^1^

SEM: standard error of the mean. * Linear-by-linear association Chi-square. ** Hyperthyroidism and diabetes mellitus. *** Depressive disorders or anxiety. ^1^ N.S.: not significant.

**Table 2 pathogens-13-00843-t002:** Perinatal complications and postnatal clinical manifestations in *C. trachomatis*-exposed newborns.

	All Newborns [n (%), n = 157]	Newborns with Positive Test for *C. trachomatis* (n = 28)	Newborns with Negative Test for *C. trachomatis* (n = 129)	*p* Value
**CHARACTERISTICS OF NEONATES—** **Number of babies (%)**				
Gender ^a^				N.S. ^1^
Male	79 (50.3)	16 (57.1)	63 (48.8)	
Female	78 (49.7)	12 (42.9)	66 (51.2)	
Born before 37 completed gestational weeks ^a^	89 (56.7)	20 (71.4)	69 (53.5)	N.S. ^1^
Growth-restricted (<10th percentile) ^a^	16 (10.2)	2 (7.1)	14 (10.9)	N.S. ^1^
Earliest preterm births ^b^				N.S. ^1^
Term	68 (43.3)	8 (28.6)	60 (46.5)	
Preterm	9 (5.7)	2 (7.1)	7 (5.4)	
Moderate to late preterm	36 (22.9)	8 (28.6)	28 (21.7)	
Very preterm	44 (28)	10 (35.7)	34 (26.4)	
Birth weight (g) ^c^	2265 [1275–3090]	1447.5 [1222.5–3100]	2484 [1360–3090]	N.S. ^1^
Birth weight Z score ^c^	0.11 [−0.93–0.98]	−0.76 [−0.99–0.81]	0.34 [−0.85–0.98]	N.S. ^1^
Birth length (cm) ^c^	45 [39–49.5]	40 [39.5–49]	46 [39–50]	N.S. ^1^
Birth length Z score ^c^	0.16 [−0.82–0.89]	−0.66 [−0.7–0.8]	0.32 [−0.54–0.82]	N.S. ^1^
Birth head circumference (cm) ^c^	31.5 [28.5–34]	30 [27.8–34]	32 [29–34]	N.S. ^1^
Birth head circumference Z score ^c^	0.14 [−0.68–0.82]	−0.27 [−0.88–0.82]	0.27 [−0.5–0.8]	N.S. ^1^
Very low birth weight ^a^	86 (54.8)	20 (71.4)	66 (51.2)	N.S. ^1^
Extremely low birth weight ^a^	54 (34.4)	16 (57.1)	38 (29.5)	0.008
Apgar score < 7 at minute ^a^	74 (47.1)	12 (42.9)	62 (48.1)	N.S. ^1^
Apgar score < 7 at five minutes ^a^	25 (15.9)	2 (7.1)	23 (17.8)	N.S. ^1^
**COMPLICATIONS—** **Number of babies (%)**				
Conjunctivitis ^a^	21 (13.4)	8 (28.6)	13 (10.1)	0.02
Perinatal pneumonia ^a^	24 (15.3)	15 (53.6)	9 (7)	<0.001
Respiratory distress syndrome ^a^	59 (37.6)	2 (7.1)	57 (44.2)	<0.001
Late onset sepsis ^a^	40 (25.5)	13 (46.4)	27 (20.9)	0.008

^a^ Number of patients and frequency (%), Fisher’s exact test *p* value. ^b^ Number of patients and frequency (%), linear-by-linear association Chi-square test *p* value. ^c^ Median [interquartile range], Mann–Whitney U test *p* value. ^1^ N.S.: not significant.

**Table 3 pathogens-13-00843-t003:** Cox proportional hazards models to estimate mother-to-child transmission risk and symptomatic neonatal infection.

Explanatory Variables	Covariates	Estimate (SE)	Hazard Ratio	95% CI	*p* Value
Mother-to-Child Transmission	Mother *C. trachomatis*-negative test	3.19 (1.06)	24.38	3.06–194.3	0.003
	Birthweight stratification (g)				<0.001
	>2500	Ref.			
	1501 to 2500	0.06	1.06	0.31–3.65	N.S. ^1^
	<1500	1.48	4.37	1.68–11.38	0.003
Perinatal pneumonia	*C. trachomatis*-infected baby	1.27 (0.45)	3.55	1.48–8.51	0.004
	Birthweight stratification (g)				<0.001
	>2500	Ref.			
	1501 to 2500	2.85	17.3	1.91–156.6	0.01
	<1500	4.43	83.7	9.95–704.5	<0.001
Respiratory Distress Syndrome	Birthweight stratification (g)				<0.001
	>2500	Ref.			
	1501 to 2500	1.82 (0.36)	6.2	3.04–12.65	<0.001
	<1500	2.99 (0.33)	19.92	10.53–37.69	<0.001
Conjunctivitis	Birthweight stratification (g)				<0.001
	>2500	Ref.			
	1501 to 2500	1.45 (0.69)	4.27	1.11–16.39	0.035
	<1500	2.91 (0.58)	18.29	5.88–56.91	<0.001
Late onset sepsis	Birthweight stratification (g)				<0.001
	>2500	Ref.			
	1501 to 2500	1.28 (1.0)	3.6	0.5–25.82	N.S. ^1^
	<1500	4.11 (0.75)	60.93	13.97–265.8	<0.001

^1^ N.S.: not significant.

**Table 4 pathogens-13-00843-t004:** Relative and attributable risk of mother-to-child transmission and symptomatic neonatal *C. trachomatis* infection by maternal exposure levels.

Symptomatic Neonatal Infection	Adjusted Relative Risk (95% CI)	Attributable Risk
Mother-to-child transmission		
Early antenatal detection	Ref.	0.98
Mid-antenatal detection	11.1 (1.4–88.1)	
Late antenatal detection	20.1 (3.0–135.1)	
*p* value	0.001	
Perinatal pneumonia		
Early antenatal detection	Ref.	0.65
Mid-antenatal detection	7.5 (3.5–16.1)	
Late antenatal detection	6.2 (3.4–11.4)	
*p* value	0.002	
Conjunctivitis		
Early antenatal detection	Ref.	0.41
Mid-antenatal detection	1.8 (0.7–4.9)	
Late antenatal detection	2.6 (1.3–5.1)	
*p* value	0.007	
Late onset sepsis		
Early antenatal detection	Ref.	0.45
Mid-antenatal detection	2.2 (0.97–5.2)	
Late antenatal detection	2.6 (1.3–5.2)	
*p* value	0.01	

*p* value calculated by the Cochran–Mantel–Haenszel method. Ref.: reference category.

## Data Availability

The datasets used and/or analyzed during the current study are available from the corresponding author upon reasonable request.

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
