# Peer review of "Identifying the Impact of Chlamydia trachomatis Screening and Treatment on Mother-to-Child Transmission, and Respiratory Neonatal Outcomes in Mexico"

_pathogens, 2024, doi:10.3390/pathogens13100843_

Round 1

Reviewer 1 Report (Previous Reviewer 1)

Comments and Suggestions for Authors

For consistency, I kept my initial comments and the authors’ responses in the report. My comments to the authors’ responses, as well as additional specific comments are presented in bold. 

General comments. The paper aimed to investigate whether a single maternal screening for Chlamydia is enough to prevent adverse pregnancy and neonatal outcomes. The authors attempted to provide data showing that there is a need for a second screening test in the third trimester, in addition to screening in early pregnancy, as maternal infection or reinfection significantly increases the risk of neonatal infection. There are a number of research gaps in our knowledge of the impact of STIs on pregnancy and neonatal outcomes, as well as mother-to-child transmission rates and optimal screening strategies therefore the study topic is of high relevance.

Comment 1: The study has serious flaws in the methodology and presentation. For the diagnosis of Chlamydia in women, an international rigorously validated test (COBAS® TaqMan CT) was used, which insures high diagnostic accuracy. On the contrary, neonatal samples were tested using an in-house PCR. Its diagnostic accuracy, which is standardly determined in comparison to some well validated test, is unknown, because the reference for this test is not in English. Then, using different tests when investigating mother-to-child transmission cannot be considered reasonable. The fact that around 30% of neonates from Chlamydia–negative mothers had neonatal Chlamydia infection could at least partly be explained by suboptimal diagnostic characteristics of the in-house test.

 Response 1: Thanks for your comments. It is important to note that, although the COBAS® TaqMan CT test has been internationally validated and results from some studies do show until 100% sensitivity and specificity, this is for single maternal screening of Chlamydia only. Our paper aimed to investigate whether a single maternal screening for Chlamydia is enough to prevent 2 adverse pregnancy and neonatal outcomes. The failure in maternal diagnosis could be related to the site of C. trachomatis colonization at the time of collected of endocervical sample, as well as to the type of C. trachomatis strain that is circulating in the region where this COBAS TaqMan CT KIT is used, low number of elementary or reticular bodies in the sample swab, inhibitors of gene amplification target, or screening test before acquiring the Chlamydial infection from her sex partner. 1. Chen, H., et al. Lactobacillus Modulates Chlamydia Infectivity and Genital Tract Pathology in vitro and in vivo. Front. Microbiol. 2022,13:877223. doi: 10.3389/fmicb.2022.877223. 2. Balakrishnan, S.N., et al. Role of Vaginal Mucosa, Host Immunity and Microbiota in Vulvovaginal Candidiasis. Pathogens 2022, 11, 618. https://doi.org/10.3390/pathogens11060618. 3. Peric, A., et al. Bacterial Colonization of the Female Upper Genital Tract. Int. J. Mol. Sci. 2019, 20(14),3405. doi: 10.3390/ijms20143405. 4. Chen, C., et al. The microbiota continuum along the female reproductive tract and its relation to uterine-related diseases. Nat. Commun. 2017, 8, 875. https://doi.org/10.1038/s41467-017-00901-0. 5. Møller JK, et al. Comparison of the Abbott RealTime CT new formulation assay with two other commercial assays for detection of wild-type and new variant strains of Chlamydia trachomatis. J Clin Microbiol. 2010;48(2):440-3. doi: 10.1128/JCM.01446-09 6. Le Roy C, et al. Swabs (dry or collected in universal transport medium) and semen can be used for the detection of Chlamydia trachomatis using the Cobas 4800 system. J Med Microbiol. 2013;62(Pt 2):217-222. doi: 10.1099/jmm.0.048652-0. Regarding the processing of neonatal samples for diagnosis of C. trachomatis, there is currently no specific commercial kit for detection of the causal agent in the newborn where bronchoalveolar aspirate/lavage and sputum samples are used. It should be noted that bronchial aspirate samples 3 from newborns show a high content of mucins and sialic acid, substances that contribute significantly to the viscosity of sputum, and that could modify the results of the PCR reaction. The negative charge of sialic acids could interact with magnesium, and therefore modify the activity of DNA polymerase which will work too quickly often making errors in the copying process, and inhibiting amplification can lead to false negative results for PCR reaction. Purification of DNA from bronchoalveolar lavage fluid respiratory samples based on cetyltrimethylammonium bromide (CTAB) has been recommended to increase the diagnostic sensitivity of PCR and confidence in negative results. Using the most widely used standard purification method for DNA extraction with phenol/chloroform followed by precipitation with cetyltrimethylammonium bromide (CTAB), the best DNA purification method has been achieved in combination with C. pneumoniae-specific real-time quantitative PCR. 1. Apfalter P, et al. Application of blood-based polymerase chain reaction for detection of Chlamydia pneumoniae in acute respiratory tract infections. Eur J Clin Microbiol Infect Dis. 2001;20(8):584-6. doi: 10.1007/s100960100554. 2. Maass M, Dalhoff K. Comparison of sample preparation methods for detection of Chlamydia pneumoniae in bronchoalveolar lavage fluid by PCR. J Clin Microbiol. 1994, 32(10):2616-9. doi: 10.1128/jcm.32.10.2616-2619.1994. Because this type of respiratory samples causes difficulties in diagnosis using the COBAS® TaqMan CT test, we have used the precipitation method with high concentrations of NaCl CTAB that precipitates polysaccharides and proteins. Some of our scientific articles published in different languages show the results of the diagnosis of neonatal C. trachomatis infection by endpoint PCR test using this purification method on sputum samples or bronchial lavage. 1. Hernandez-Trejo M, Herrera-Gonzalez NE, Escobedo-Guerra MR, de Haro-Cruz MJ, Moreno-Verduzco ER, Lopez-Hurtado M, et al. Reporting detection of Chlamydia 4 trachomatis DNA in tissues of neonatal death cases. J. Pediatr. (Rio J) 2014, 90:182- 189. 2. Marcela López-Hurtado, Gabriel Arteaga-Troncoso, Irma E. Sosa-González, Maria de Jesus de Haro-Cruz, Veronica R. Flores-Salazar & Fernando Martín Guerra-Infante. Eosinophilia in Preterm Born Infants Infected with Chlamydia trachomatis, Fetal and Pediatric Pathology 2016, 35:3, 149-158, DOI: 10.3109/15513815.2016.1153175. 3. Melissa D. González-Fernández, Marco A. Escarcega-Tame, Marcela López-Hurtado, Verónica R. Flores-Salazar, Marcos R. Escobedo-Guerra, Silvia Giono-Cerezo, Fernando M. Guerra-Infante. Identification of Chlamydia trachomatis genotypes in newborns with respiratory distres. Anales de Pediatría (English Edition) 2023, 98 (6):436-445. https://doi.org/10.1016/j.anpede.2023.04.010. Our research group has been working for a long time with the PCR technique for the detection of C. trachomatis in bronchial aspirate samples, and the results of this test have been validated simultaneously with the detection of anti-Chlamydia IgM antibodies in serum samples of newborns with respiratory distres (reference 25). Finally, it is worth mentioning that in all endpoint PCR runs of this study, controls are added, both the negative (DNA from McCoy Cells not infected with Chlamydia) and the positive (DNA from Chlamydia trachomatis serotype D, ATCC VR885D) which reduces the possibility of false positive reactions and confirms that they have the same electrophoretic shift.

Comment to response 1: I do see the authors’ points. There is no objection against using an in-house PCR assay, however, any diagnostic test, commercial or in-house, prior to its introduction needs comprehensive validation, which is commonly performed via comparison with a well-known/well validated test(s), with reporting at least diagnostic sensitivity and specificity of the test. In the referred sources, I did not find this key information.

Comment 2. Another serious concern is connected to concomitant maternal infections. The purpose of testing for HIV, HPV, vaginal infections is not clearly presented. Tests for HIV and HPV (which types?) are not described. Not clear what authors mean by “vaginal infections”, as mere presence of opportunistic bacteria like Gardnerella in the vagina cannot be considered an infection. 5 Furthermore, interpretation of results of tests for “vaginal infections”, treatment indications, treatments are not presented. There is a reference, but it is not in English.

Response 2. Indeed, the mere detection of Gardnerella vaginalis does not imply a vaginal infection because the microorganism normally colonizes the vaginal tissue without causing any type of symptoms. However, when it multiplies excessively, modifying the vaginal microflora, it can cause the infection called bacterial vaginosis. The cases of women with vaginal infection in our study were further defined by clinically identified grayish white discharge and foul odor. -L111-117: For better understanding, the sentence has been rephrased, thank you.

Comment to response 2: The part of the manuscript regarding vaginal infections is still entangled and poorly structured. In Methods (section 2.3), the authors described how they defined vaginal infection, but the Results section gives no analysis on this issue. Then, figure 2 shows no combinations of C. trachomatis with other vaginal infections, which is in discordance with its significant association with G. vaginalis detected in the study. Furthermore, the authors state that “C. trachomatis infection associated with other STIs showed a higher risk of infection for HPV” (Lines 270-271). Which STIs are meant? U. urealyticum, G. vaginalis, and Group B streptococcus (GBS) are not STIs. The part of the manuscript regarding vaginal infections should be revised, to make it clear, concise and relevant.

Comment 3. The results presented in the tables are very confusing. For example, at glance, according to Table 1, the positivity for Chlamydia is much higher in non-teenagers than in teenagers, but this is not the case, according to the text, and it needed me some efforts to understand, how it should be presented correctly. Furthermore, according to Table 2, respiratory distress syndrome was significantly more frequent in neonates negative for Chlamydia, which does not seem to be the case. All tables should be revised and corrected to be clear, unambiguously interpreted and totally consistent with the text.

Response 3. Yes, all the tables were reviewed and corrected in the text, thank you.

 Comment to response 3. The tables are much clearer now.

 Comment 4. Reference list: there are several sources in non-English language; these should not be used, especially when there is no comprehensive English abstract. What is more, around 75% of all literature sources are older than 5 years. The list should be updated, and recent most relevant studies should be cited.

Response 4. Our manuscript does not show publication bias that can be avoidable in any scientific study, where some relevant articles written in other languages may be overlooked, since only studies written in English are included. Diagnostics play a critical role in providing equitable access to universal healthcare. Maternal C. trachomatis infection has highlighted the urgency of implementing available, accessible and affordable diagnostics to prevent transmission of infection from mother to her baby worldwide. I hope you understand the deep-rooted global inequalities in global public health. Although most literature sources are more than 5 years old, this seems to reflect that the problem of mother-to-child transmission has been resolved in the United States and in certain places, but not in most countries, mainly those with limitation resources. Recently, Adachi et al. (2021) mentions that only 13 of 15 studies observe some degree of support for prenatal chlamydia screening and treatment interventions that can lead to a decrease in adverse pregnancy and newborn outcomes (reference . This notable lack of studies highlights the need for more up-to-date research on mother-to-child transmission of C. trachomatis, particularly in lowand middle-income settings (Adachi KN, Nielsen-Saines K and Klausner JD. Chlamydia trachomatis Screening and Treatment in Pregnancy to Reduce Adverse Pregnancy and Neonatal Outcomes: A Review. Front. Public Health 2021,9:531073. doi: 10.3389/fpubh.2021.531073). The above also leads us to consider the greater productivity associated with research projects that solve medical problems in countries with high economic resources in contrast to the low productivity shown by countries with few resources. I only hope that our scientific article is published in the open access journal “Pathogens” to address the deeply rooted global inequalities in public health and allow  impactful research to be visible and accessible to all health professionals, planners of health policies and the general public without barriers or economic biases. Our research aimed to highlight the importance of maternal diagnosis and treatment of Chlamydia trachomatis in the context of global health, including COBAS TaqMan CT with the urgent need for dual screening as suggested in the CDC Guidelines, for the deployment of diagnostics at maternal health care points around the world including low- and middle-income countries; as well as to influence health policies and regulation of diagnoses in maternal health where, like Mexico, there are problems with the supply chain of medicines, affordability, accessibility and availability of essential diagnoses after the COVID-19 pandemic.

 Comment to response 4. I am very well aware of inequalities in global public health, as well as of the importance of meaningful data from all over the world. However, it must be assured that readers are able to understand the study background and methodology in full, for which using updated research data, presented in an internationally used language, is essential.   

 Additional specific comments

1. Line 117-118: The authors state that testing for vaginal infections was performed routinely and refer to a paper, which is a review paper for BV, giving no information how in fact testing for vaginal infections was performed.

2.     Line 120-125: Is this a routine practice that pregnant women with a history of HPV infection are referred for colposcopy and biopsies of the affected tissue for histopathology HPV testing? Or was it done only for the study? Why biopsy samples, not endocervical swabs, were tested? Furthermore, if the authors aimed to investigate an association between C. trachomatis and HPV, all study subjects should have been tested.   

3.     Lines 162-166: It is stated that “the following samples were obtained from each baby: …“, but tracheal lavage and conjunctival discharge were only taken in specific cases.

4. Line 163: Not clear what “isolation of atypical microorganisms” means and if rhinopharyngeal secretion were tested for C. trachomatis, please specify.

5.     Line 202: “Diagnosing intrauterine C. trachomatis infection” should be replaced with “Diagnosing C. trachomatis infection”

6.     Table 1. Not clear what “Endocrinological involvement” and “Psychiatric disorders” are, please specify.

7.     Lines 208-209: The authors state that “The sensitivity and specificity of the COBAS® TaqMan CT Test on clinical specimens are 93% and 99.3%, respectively [25]”. However, the referred source mentions COBAS Amplicor, not COBAS® TaqMan CT test, with the sensitivity and specificity of 100% and 99.8%, respectively, for cervicovaginal samples. Please clarify.

8.     Line 276: HPV 6 and 71 are not high-risk genotypes, please correct

Author Response

Response to Reviewer 1 Comments

Comment to response 1: I do see the authors’ points. There is no objection against using an in-house PCR assay, however, any diagnostic test, commercial or in-house, prior to its introduction needs comprehensive validation, which is commonly performed via comparison with a well-known/well validated test(s), with reporting at least diagnostic sensitivity and specificity of the test. In the referred sources, I did not find this key information.

Response 1: Thanks for your comments. The point was corrected in the text.

L220-223: The sensitivity and specificity of in-house PCR assay calculated for diagnosing C. trachomatis in blood sample from newborns are 94.4 % and 88.5%, while bronchoalveolar lavage samples 27.8% and 92.3%, respectively [27].

Comment to response 2: The part of the manuscript regarding vaginal infections is still entangled and poorly structured. In Methods (section 2.3), the authors described how they defined vaginal infection, but the Results section gives no analysis on this issue. Then, figure 2 shows no combinations of C. trachomatis with other vaginal infections, which is in discordance with its significant association with G. vaginalis detected in the study. Furthermore, the authors state that “C. trachomatis infection associated with other STIs showed a higher risk of infection for HPV” (Lines 270-271). Which STIs are meant? U. urealyticum, G. vaginalis, and Group B streptococcus (GBS) are not STIs. The part of the manuscript regarding vaginal infections should be revised, to make it clear, concise and relevant.

Response 2: Thanks for your comments. All points were corrected in the text

L260-263: Sixty-six women (10.5%) tested positive for C. trachomatis as one of several vaginal infections, and of these, 53 women (8.4%) were C. trachomatis positive only in endocervical specimens (all of them were subsequently treated with azithromycin).

Additional specific comments

  1. Line 117-118: The authors state that testing for vaginal infections was performed routinely and refer to a paper, which is a review paper for BV, giving no information how in fact testing for vaginal infections was performed.

L118-119: Laboratory diagnosis of vaginal infections was performed routinely and described elsewhere [19].

Sharon L Hillier, Michele Austin, Ingrid Macio, Leslie A Meyn, David Badway, Richard Beigi. Diagnosis and Treatment of Vaginal Discharge Syndromes in Community Practice Settings. Clin Infect Dis. 2021 May 4;72(9):1538-1543. doi: 10.1093/cid/ciaa260.

  1. Line 120-125: Is this a routine practice that pregnant women with a history of HPV infection are referred for colposcopy and biopsies of the affected tissue for histopathology HPV testing? Or was it done only for the study? Why biopsy samples, not endocervical swabs, were tested? Furthermore, if the authors aimed to investigate an association between C. trachomatis and HPV, all study subjects should have been tested.

R= Routinely all patients with a history of HPV infection undergo cervical cytology, if it is positive, colposcopy is performed and if there are abnormal cellular changes, a biopsy is taken and, depending on the histopathological result, follow-up is followed until the pregnancy is resolved.

Thirty-one patients with a history of HPV infection were evaluated and it was highlighted which had concomitant infection.

www.asccp.org/consensus2012

  1. Lines 162-166: It is stated that “the following samples were obtained from each baby: …“, but tracheal lavage and conjunctival discharge were only taken in specific cases.

R: Yes, to increase the diagnostic sensitivity of PCR and the confidence in the results of truly negative babies in cases of respiratory infections.

  1. Line 163: Not clear what “isolation of atypical microorganisms” means and if rhinopharyngeal secretion were tested for C. trachomatis, please specify.

R: Atypical microorganisms are considered bacteria that are not stained with Gram techniques and are challenging to grow in an artificial medium, such as Mycoplasma, Ureaplasma, and Chlamydia trachomatis.

  1. Line 202: “Diagnosing intrauterine C. trachomatis infection” should be replaced with “Diagnosing C. trachomatis infection”.

-L203: The phrase has been replaced, thank you.

  1. Table 1. Not clear what “Endocrinological involvement” and “Psychiatric disorders” are, please specify.

-L274-275: ** Hyperthyroidism and diabetes mellitus. *** Depressive disorders or anxiety.

  1. Lines 208-209: The authors state that “The sensitivity and specificity of the COBAS® TaqMan CT Test on clinical specimens are 93% and 99.3%, respectively [25]”. However, the referred source mentions COBAS Amplicor, not COBAS® TaqMan CT test, with the sensitivity and specificity of 100% and 99.8%, respectively, for cervicovaginal samples. Please clarify.

-L210: Yes, the cite was corrected, thank you.

  1. Line 276: HPV 6 and 71 are not high-risk genotypes, please correct.

-L286: Yes, the HPV genotypes were corrected, thank you.

Reviewer 2 Report (Previous Reviewer 2)

Comments and Suggestions for Authors

This version of the manuscript has been improved significantly from the previous version and I am satisfied with the revisions made per previous comments. 

Author Response

Response to Reviewer 2 Comments

This version of the manuscript has been improved significantly from the previous version and I am satisfied with the revisions made per previous comments.

Thank you for your kind comments.

Reviewer 3 Report (Previous Reviewer 4)

Comments and Suggestions for Authors

Authors use Chlamydia instead of C. trachomatis in lines: 29, 30, 67, 87, 143, 149, 213, 218, 231,247, 306, 316, 318, 325, 332, 353, 356, 362, 363, 364, 375, 378, 386, 392, 396, 399, 400, 401, 402, 407, 420, 424, 426, 428, 432, 441, 445, 453, 464, 466, 469, 471, 473, 476, 480, 482, 491, 500, 502, 506, 511, 514, 526, 528, Figure 3, Figure 4 legend, Table 3, Table 4 legend. This must be corrected.

In the abstract “evaluated to question” ? What do authors mean?

Please erase “recent” from line 52. A study published in 2016 is not recent.

At Ln 56, the reference 5 must be cited again.

Ln 76 erase “already”

Lines 81 to 83, it is not clear what do authors want to say. Are they advocating the benefits of case-cohort studies?

Lines 120 to 138, are useless. HPV infection has no direct relation with the study so it is superfluous.

Line 173: “Vaginal infection”? Do authors mean vaginitis?

Line 202, “diagnosing intrauterine”. No. Authors diagnose C. trachomatis infection at the cervix, not intrauterine.

Line 206, omp1 must be in italic, it is a gene name. And authors must choose ompA, most correct, as stated in line 219, but ompA must be in italic

Line 270 “other STIs” ? Which ones?

Lines 285 to 293, evaluation of commensal mycoplasma for what reason? What is the interest?

Line 293, “2.7%” It represents how many individuals? N=? The same N is missing for the 42.9% in Line 426

I do not see the interest of table 3. And what do authors mean, in this table and in table 4, by “Ref”?

Discussion is exaggeratedly long and repetitive

Section 5. is pointless

Conclusions: if C. trachomatis infection is not detected in mothers, they will not be treated. It is wrong to state that they are “inadequately treated”

Comments on the Quality of English Language

No further comments 

Author Response

Response to Reviewer 3 Comments

Comments and Suggestions for Authors

Authors use Chlamydia instead of C. trachomatis in lines: 29, 30, 67, 87, 143, 149, 213, 218, 231,247, 306, 316, 318, 325, 332, 353, 356, 362, 363, 364, 375, 378, 386, 392, 396, 399, 400, 401, 402, 407, 420, 424, 426, 428, 432, 441, 445, 453, 464, 466, 469, 471, 473, 476, 480, 482, 491, 500, 502, 506, 511, 514, 526, 528, Figure 3, Figure 4 legend, Table 3, Table 4 legend. This must be corrected.

Responses:

Yes, all points were corrected in the text, thank you.

In the abstract “evaluated to question” ? What do authors mean?

-L25 it was erase “to question”

Please erase “recent” from line 52. A study published in 2016 is not recent.

-L52 it was erase “recent”

At Ln 56, the reference 5 must be cited again.

-L56 it was referenced the cite 5.

Ln 76 erase “already”

-L76 it was erase “already”

Lines 81 to 83, it is not clear what do authors want to say. Are they advocating the benefits of case-cohort studies?

-L82: For better understanding, the sentence has been rephrased, thank you.

Lines 120 to 138, are useless. HPV infection has no direct relation with the study so it is superfluous.

-L120 – 138: The paragraph was retained at the request of a reviewer.

Line 173: “Vaginal infection”? Do authors mean vaginitis?

-L173: Vaginitis, thank you.

Line 202, “diagnosing intrauterine”. No. Authors diagnose C. trachomatis infection at the cervix, not intrauterine.

-L203: For better understanding, the sentence has been rephrased, thank you.

Line 206, omp1 must be in italic, it is a gene name. And authors must choose ompA, most correct, as stated in line 219, but ompA must be in italic

-L206 and –L219: The points were corrected in the text, thank you.

Line 270 “other STIs” ? Which ones?

L262: Sorry. The word “STIs” has been replaced by “Infecciones vaginales”, thank you.

Lines 285 to 293, evaluation of commensal mycoplasma for what reason? What is the interest?

R: A possible confounding factor associated with perinatal complications would be the presence of U. urealyticum/M. hominis. The most frequently involved infectious agent associated with preterm delivery and chorioamnionitis.

Line 293, “2.7%” It represents how many individuals? N=? The same N is missing for the 42.9% in Line 426

-L300-301: The number of women by period of gestation was included in the text, thank you.

I do not see the interest of table 3. And what do authors mean, in this table and in table 4, by “Ref”?

Regarding table 3, We aimed to calculate the probability of vertical transmission of C. trachomatis estimated from clinical study. To calculate a reliable estimate for the risk of vertical transmission on the perinatal pneumonia, RDS, conjunctivitis, and late-onset sepsis adjusted for maternal age, smoking, alcohol consumption, and sex of newborn.

Regarding table 4, We aimed to adjusted estimates of the relative and attributable risks of mother–to–child transmission and symptomatic infection from C. trachomatis–positive newborns by maternal screening categories during pregnancy. The attributable risk, expressed as a population attributable fraction, was calculated to assess the portion of disease burden preventable by avoiding exposure with adequately diagnosed.

L396: Ref.: Reference category.

Discussion is exaggeratedly long and repetitive

Section 5. is pointless

-L 554 - 560 were removed, but section 5 was retained at the request of a reviewer.

Conclusions: if C. trachomatis infection is not detected in mothers, they will not be treated. It is wrong to state that they are “inadequately treated”

-L555 - 557: For better understanding, the sentence has been rephrased, thank you.

Round 2

Reviewer 1 Report (Previous Reviewer 1)

Comments and Suggestions for Authors

The authors mostly addressed the reviewer' concerns

Author Response

Response to Reviewer 1 Comments

Comment to response 1: The authors mostly addressed the reviewer' concerns.

Response 1: Thank you very much for your comments.

We sincerely appreciate your insightful comments, which have significantly contributed to the refinement of our research. Your input was invaluable to us.

Reviewer 3 Report (Previous Reviewer 4)

Comments and Suggestions for Authors

For Authors:

Line 277 “other STIs” ? Which ones?

What is PROM? Prenatal rupture of membranes? It is not stated.

HPV, HIV, urogenital mycoplasma and Ureaplasma, bacterial vaginites/vaginosis results are useless

Author Response

Response to Reviewer 3 Comments

Line 277 “other STIs” ? Which ones?

-L278: For better understanding, the sentence has been rephrased, thank you.

What is PROM? Prenatal rupture of membranes? It is not stated.

-L265: Premature rupture of membrane (PROM) is stated.

HPV, HIV, urogenital mycoplasma and Ureaplasma, bacterial vaginites/vaginosis results are useless.

R: The results were retained at a reviewer's request, as they are crucial in understanding how new methodologies can be created in the investigation of the mother-child transmission of C. trachomatis. This will help in knowing pathological relationships and make possible an idea of medical progress. On the other hand, these results, particularly the cost-effectiveness of screening for C. trachomatis infection and the impact of inadequate detection in the mother and the risk of disease in the baby, will always be a discussion point; however, in this investigation, the cost-effectiveness of screening for C. trachomatis infection was correctly substantiated and played a significant role in our research. The recommendation outlined in this study highlights the need to carefully weigh the risks and benefits of early detection and retesting of women with vaginal C. trachomatis infection in pregnancy.

We sincerely appreciate your insightful comments, which have significantly contributed to the refinement of our research. Your input was invaluable to us.

This manuscript is a resubmission of an earlier submission. The following is a list of the peer review reports and author responses from that submission.

Round 1

Reviewer 1 Report

Comments and Suggestions for Authors

The paper aimed to investigate whether a single maternal screening for Chlamydia is enough to prevent adverse pregnancy and neonatal outcomes. The authors attempted to provide data showing that there is a need for a second screening test in the third trimester, in addition to screening in early pregnancy, as maternal infection or reinfection significantly increases the risk of neonatal infection. There is a number of research gaps in our knowledge of the impact of STIs on pregnancy and neonatal outcomes, as well as mother-to-child transmission rates and optimal screening strategies, therefore the study topic is of high relevance.

However, the study has serious flaws in methodology and presentation. For the diagnosis of Chlamydia in women, an international rigorously validated test (COBAS® TaqMan CT) was used, which insures high diagnostic accuracy. On the contrary, neonatal samples were tested using an in-house PCR. Its diagnostic accuracy, which is standardly determined in comparison to some well-validated test, is unknown, because the reference for this test is not in English. Then, using different tests when investigating mother-to-child transmission cannot be considered reasonable. The fact that around 30% of neonates from Chlamydia–negative mothers had neonatal Chlamydia infection could at least partly be explained by suboptimal diagnostic characteristics of the in-house test. 

Another serious concern is connected to concomitant maternal infections. The purpose of testing for HIV, HPV, vaginal infections is not clearly presented. Tests for HIV and HPV (which types?) are not described. Not clear what authors mean by “vaginal infections”, as mere presence of opportunistic bacteria like Gardnerella in the vagina cannot be considered an infection. Furthermore, interpretation of results of tests for “vaginal infections”, treatment indications, treatments are not presented. There is a reference, but it is not in English.  

The results presented in the tables are very confusing. For example, at glance, according to Table 1, the positivity for Chlamydia is much higher in non-teenagers than in teenagers, but this is not the case, according to the text, and it needed me some efforts to understand, how it should be presented correctly.  Furthermore, according to Table 2, respiratory distress syndrome was significantly more frequent in neonates negative for Chlamydia, which does not seem to be the case. All tables should be revised and corrected to be clear, unambiguously interpreted and totally consistent with the text.  

Reference list: there are several sources in non-English language; these should not be used, especially when there is no comprehensive English abstract. What is more, around 75% of all literature sources are older than 5 years. The list should be updated, and recent most relevant studies should be cited.

Reviewer 2 Report

Comments and Suggestions for Authors

It was a well thought study which describes the impact of Chlamydia trachomatis screening and treatment on mother-to child transmission but poorly executed because authors used a different method for the detection of Chlamydia from mothers and babies, therefore the drawn conclusion are not supported.  

Specific comments are given in file as well

Reviewer 3 Report

Comments and Suggestions for Authors

Chlamydia trachomatis infection has been associated with adverse pregnancy and neonatal outcomes such as premature rupture of membranes, premature delivery, low birth weight, conjunctivitis, and pneumonia in infants, so monitoring is important. Their work is praiseworthy and deserves all the credit.

General comments

The introduction is concrete, and the objective of the work is well justified. Overall, the study is well-designed and presented in a good way.

It is recommended to make some changes to the English language, some typos are found.

Note: I didn´t find any supplementary material.

Minor revisions

I recommend using institutional e-mails.

I recommended adding a graphical Abstract.

In the Study Design section, the PRISMA Checklist statement is highly recommended (http://prisma-statement.org/PRISMAStatement/Checklist ).

Figure 1 must be improved. Sometimes the complete information is not displayed. I highly recommend using the PRISMA Flow Diagram for Study Selection or the PRISMA IPD Flow Diagram http://prisma-statement.org/Extensions/IndividualPatientData .

Lines 109 – 110: Vaginal swabs ……. 1 g oral dose of azithromycin.; ¿Are there any references?

Lines 110 – 113: The test…… detected RNA and DNA, respectively. ¿Are there any references?

Line 150: What means US?

Please homogenize the tables, sometimes p-value is displayed, and sometimes P-value is displayed.

It is also necessary to adjust the disposition of the results within the tables, sometimes it is confusing to know which is the corresponding result.

It is necessary to reorganize the results section. As mentioned in the journal's guidelines, tables or figures should appear as they are mentioned in the text.

In the results section, authors could add a bar or pie chart showing the incidence of the various factors or conditions linked to Chlamydia infection, as well as the ages of the mothers and the significant differences between these groups.  You could also add a graph of the incidence of post-birth illnesses. A graph would enhance the work and would be easier for the reader to understand.

Since the study was conducted in 2018, were patients followed up or monitored? Both moms and/or babies.

Could you add a section after the conclusions of Perspectives on post-pregnancy studies, or during the quarantine stage?

In the reference section, scientific names should be in italics.

Comments on the Quality of English Language

Some typos are found.

Reviewer 4 Report

Comments and Suggestions for Authors

Authors intended to determine whether a single Chlamydia trachomatis test would be enough to prevent mother to child transmission of C. trachomatis.

Authors must replace “Chlamydia” by “C. trachomatis” in the whole text, including figures and tables, because this is the species under study. Authors could have evaluated the impact of other chlamydial infection, namely Chlamydia pneumonia, a common cause of respiratory infection, but this was not the case. Authors must address this limitation in the discussion.

Authors must put in italic the name of bacteria, something that has not been done in some parts of the text, namely in pages 5 and 6.

In the introduction, please replace the WHO document used by the most recent, 2023, and update number of estimates.

Ln 46, erase “persistent” and replace by ‘common’

Figure 1 has been incorrectly uploaded so much of the text is missing.

Ln 131, what is “m controls” ?

Methodology is a problem in this paper because different tests, including different methods for DNA extraction have been used in mothers samples and in neonates samples. The in house method comprehending one single target in ompA should be less sensitive than the commercial method with two targets, which is largely recognized by its sensitivity. Authors should fear the lack of sensitivity and specificity of the in house technique. It is not understandable the lack of standardization in the sampling of neonates. Why were they not all sampled in the nasopharynx? Why taking so many blood samples which are not adequate for C. trachomatis diagnosis?

Ln 256, use a single decimal place instead of 1.52%

Table 1, erase “other STI infections”, authors could have evaluated Neisseria gonorrhoeae, Trichomonas vaginalis, Mycoplasma genitalium in parallel, as they didn’t this subtitle must disappear and authors must discuss this limitation in the discussion.. Considering the low risk of the population under study, Mycoplasma hominis and Ureaplasma urealyticum and U. parvum could/should have been evaluated. As they wasn’t authors must also address this limitation in the discussion.

In the discussion please erase “this study was the first to document the…” there were so many evaluations regarding mother to child transmission of C. trachomatis in the past that it sounds overweening.

The definition of NAAT in the discussion should be avoided, it should be included in the materails and methods when describing COBAS.

Authors consider more probable that the COBAS test failed to detect infected mothers than the risk of them getting infected after being tested. This should be avoided because there was no questionnaire concerning sexual life of pregnant women. Until what phase of the pregnancy they had sex? Did they have new partners? Did their partners had new partners?

The whole text regarding the Swedish variant is a bit useless, and should be discarded, because it happened before Cobas included two targets. The probability of appearing clones presenting mutations involving simultaneously should be lower than the probability of false positives using an in-house technique.

Ln 428 replace “can” by ‘could hipothetically’